# *O*-GlcNAcylation and Regulation of Galectin-3 in Extraembryonic Endoderm Differentiation

**DOI:** 10.3390/biom12050623

**Published:** 2022-04-22

**Authors:** Mohamed I. Gatie, Danielle M. Spice, Amritpal Garha, Adam McTague, Mariam Ahmer, Alexander V. Timoshenko, Gregory M. Kelly

**Affiliations:** 1Department of Biology, Western University, London, ON N6A 5B7, Canada; mgatie@uwo.ca (M.I.G.); dspice@uwo.ca (D.M.S.); agarha@uwo.ca (A.G.); amctagu@uwo.ca (A.M.); mahmer@uwo.ca (M.A.); atimoshe@uwo.ca (A.V.T.); 2Collaborative Specialization in Developmental Biology, Western University, London, ON N6A 5B7, Canada; 3Department of Physiology and Pharmacology, Western University, London, ON N6A 5B7, Canada; 4Lawson Health Research Institute, London, ON N6A 5B7, Canada; 5Children’s Health Research Institute, London, ON N6A 5B7, Canada

**Keywords:** *O*-GlcNAc, galectin-3, unconventional secretion, extraembryonic endoderm, cell differentiation

## Abstract

The regulation of proteins through the addition and removal of *O*-linked β-*N*-acetylglucosamine (*O*-GlcNAc) plays a role in many signaling events, specifically in stem cell pluripotency and the regulation of differentiation. However, these post-translational modifications have not been explored in extraembryonic endoderm (XEN) differentiation. Of the plethora of proteins regulated through *O*-GlcNAc, we explored galectin-3 as a candidate protein known to have various intracellular and extracellular functions. Based on other studies, we predicted a reduction in global *O*-GlcNAcylation levels and a distinct galectin expression profile in XEN cells relative to embryonic stem (ES) cells. By conducting dot blot analysis, XEN cells had decreased levels of global *O*-GlcNAc than ES cells, which reflected a disbalance in the expression of genes encoding *O*-GlcNAc cycle enzymes. Immunoassays (Western blot and ELISA) revealed that although XEN cells (low *O*-GlcNAc) had lower concentrations of both intracellular and extracellular galectin-3 than ES cells (high *O*-GlcNAc), the relative secretion of galectin-3 was significantly increased by XEN cells. Inducing ES cells toward XEN in the presence of an *O*-GlcNAcase inhibitor was not sufficient to inhibit XEN differentiation. However, global *O*-GlcNAcylation was found to decrease in differentiated cells and the extracellular localization of galectin-3 accompanies these changes. Inhibiting global *O*-GlcNAcylation status does not, however, impact pluripotency and the ability of ES cells to differentiate to the XEN lineage.

## 1. Introduction

Many intracellular proteins are post-translationally modified by the addition of β-*N*-acetylglucosamine (GlcNAc) to the hydroxyl moiety of serine or threonine residues [1,2]. This *O*-GlcNAcylation acts in competition with phosphorylation at the same or nearby amino acids [3,4,5], demonstrating its integral function in protein regulation. Unlike protein phosphorylation and dephosphorylation, which are carried out by a variety of kinases and phosphatases, respectively, *O*-GlcNAcylation is regulated by only two enzymes: *O*-GlcNAc transferase (OGT), which transfers GlcNAc to proteins from a donor UDP-GlcNAc derived from the hexosamine biosynthesis pathway [6], and N-acetyl-β-D-glucosaminidase (*O*-GlcNAcase, OGA), which removes GlcNAc from proteins [7].

*O*-GlcNAcylation is a dynamic process that is regulated by multiple mechanisms. As UDP-GlcNAc is derived through the hexosamine biosynthesis pathway, the availability of GlcNAc is highly regulated by glucose metabolism [8,9] and insulin [10]. *O*-GlcNAcylation is also sensitive to free fatty acids [11] and thermal and oxidative stress [12,13,14]. The addition or removal of *O*-GlcNAc in response to nutrients and stress is not universal as the *O*-GlcNAcylated proteome is dynamic, and evidence exists showing many proteins with increased *O*-GlcNAcylation, while others have less *O*-GlcNAc [12]. *O*-GlcNAcylation is also differentially regulated during development and differentiation. The high levels of *O*-GlcNAc on specific proteins play important roles in maintaining pluripotency in mouse embryonic stem (ES) cells as it can regulate the activity of OCT4 and SOX2 [15] or inhibit the differentiation of ectoderm in human ES cells [16]. In contrast, global *O*-GlcNAcylation decreases in response to the induced differentiation of neutrophils [17] and cardiomyocytes [18].

Galectins is a family of proteins that binds β-galactoside-containing glycans and confers the ability to influence apoptosis, autophagy, transcriptional regulation, Wnt signaling and cell division, or interact with soluble and membrane-bound glycoproteins in the extracellular milieux [19]. Changes in the expression of multiple galectin family members occur in response to a variety of stress conditions [20]. Galectins are also involved with the differentiation of many cell lineages, which has been proposed to rely on *O*-GlcNAc-dependent mechanisms controlling the secretion and localization of galectins in cells [21]. In fact, while intracellular galectin-3 is *O*-GlcNAcylated, a preferentially deglycosylated protein is secreted from transfected HeLa cells [22]. Both *O*-GlcNAcylation of galectins and interaction of galectins with *O*-GlcNAcylated effector molecules are considered as potential molecular patterns in the context of cellular differentiation and proliferation [21,22,23,24]. However, little is known about the role of galectins and *O*-GlcNAc in extraembryonic endoderm (XEN) formation. XEN cells differentiate from the inner embryonic cell mass in mice at E4.5, and they are essential for later patterning and segmentation [25]. We have demonstrated a role of oxidative molecules in the differentiation of the F9 teratocarcinoma cell line, a model of XEN differentiation [26,27], and although much has been reported on their formation, there are few studies directly investigating the role of *O*-GlcNAc in this lineage [28,29].

To address this shortcoming, the goal of this study was to understand the role of *O*-GlcNAcylation, and specifically the secretion of *O*-GlcNAc-sensitive galectin-3 in processes of XEN differentiation. Toward that end, we used the relevant models of mouse embryonic and teratocarcinoma cells to demonstrate that *O*-GlcNAc levels decreased in response to XEN differentiation and that galectin-3 altered its intracellular to extracellular localization depending on cellular *O*-GlcNAc homeostasis. We showed that the lowering global *O*-GlcNAc levels in XEN cells led to the secretion of galectin-3 and had no effects on pluripotency.

## 2. Materials and Methods

### 2.1. Cell Culture

Three different mouse cell lines were used in this study representing ES cells (E14TG2a, referred to as E14), extraembryonic endoderm XEN cells (known as E4 cells), and F9 embryonal carcinoma cells inducible to endoderm differentiation by all-trans retinoic acid.

Mouse embryonic stem cell line ES-E14TG2a (E14) was obtained from the University of California, Davis, and they were maintained on 0.1% gelatin-coated tissue culture dishes as previously described [30]. Briefly, ES cells were cultured in 50/50 KnockOut™ DMEM/F-12 and neurobasal™ medium (Gibco) supplemented with N-2™, vitamin A-free B-27™ (Gibco), 1 μM MEK inhibitor PD0325901, and 3 μM GSK3 inhibitor CHIR99021 (both from APExBIO), and 1000 U/mL LIF (Millipore-Sigma, Burlington, MA, USA). E4 mouse extraembryonic endoderm [31] cells were maintained on 0.1% gelatin-coated tissue culture dishes in RPMI 1640 medium (cat. # R5886-500 mL) supplemented with 15% fetal bovine serum (FBS, cat.# 12483020) (both from ThermoFisherScientific). Both media were supplemented with GlutaMAX™ and 2-mercaptoethanol (cat.# 21985023, ThermoFisherScientific, Waltham, MA, USA), and all cells were maintained at 37 °C and 5% CO_2_ and assessed for mycoplasma and karyotypic abnormalities at the beginning of the project. The chemical induction of E14 cells toward XEN lineage was performed as previously reported [32]. Briefly, ES cells were seeded onto 0.1% gelatin-coated tissue culture plates in XEN medium (RPMI1640 media, 0.5X B-27 Supplement without insulin, 2 mM GlutaMAX™, and 0.1 mM 2-mercaptoethanol) for 2 days. Base XEN medium was supplemented with 3 µM CHIR99021 and 20 ng/mL activin A (R&D Systems), and the induction medium was changed every 2 days for 10 days.

F9 mouse teratocarcinoma cells (ATCC, cat. #CRL-1720) were cultured in DMEM (Corning, cat. # 10-013-CV), supplemented with 10% FBS and 1% penicillin-streptomycin (Gibco), and cultured on adherent tissue culture dishes (Sarstedt). For primitive endoderm differentiation, cells were treated with 0.1 μM all-trans retinoic acid (Millipore-Sigma, cat. # R2625) or vehicle control, dimethyl sulfoxide (DMSO; BioShop), for 4 days. To differentiate cells to the parietal endoderm, F9 cells were treated with 0.1 μM all-trans retinoic acid, and after two days, they were subsequently treated with 1 mM of dibutyryl-cAMP (Millipore-Sigma, cat.# D0627) for 2 days. Cells were cultured at 37 °C and 5% CO_2_ and passaged every 4 days or when cultures reached 70% confluency.

### 2.2. Western and Immunodot Blot Assays

Cells were lysed in RIPA buffer (150 mM NaCl, 1% Triton-X-100, 0.5% deoxycholate, 0.1% SDS, and 50 mM Tris, pH 8.0, with 1:100 Halt protease inhibitor cocktail from ThermoFisherScientific, cat. # PI87786) before being quantified by a *DC* Protein Assay (Bio-Rad, cat. # 5000116, Hercules, CA, USA). Approximately 10–30 μg of proteins was added to the 5X SDS loading buffer (30% glycerol, 0.02% bromphenol blue, 10% SDS, 250 mM Tris, pH 6.8) containing 10% 2-mercaptoethanol. Proteins were separated by electrophoresis on 10% polyacrylamide gels for 1.5 h at 120 V at room temperature. After separation, proteins were transferred overnight at 20 V at 4 °C to Western blot PVDF membranes (Bio-Rad, cat. # 1620177) in 20% methanol/Tris-glycine transfer buffer. Following transfer, membranes were washed in TBS-T (19.8 mM Tris, 150 mM NaCl, 0.1% Tween-20, pH 7.6), and then they were incubated in blocking solution (5% skim milk in TBS-T) for 2 h at room temperature. Membranes were then washed in TBS-T before incubation overnight at 4 °C with primary antibody solution containing 5% bovine serum albumin (BSA) in TBS-T. Primary antibodies were mouse monoclonal pan-specific antibody to *O-*GlcNAc, RL2 (1:1000; Thermo Scientific, cat. # MA1-072), rabbit anti-galectin-3 (1:1000, Santa Cruz Biotechnology, cat. # sc-20157), and mouse anti-β-actin (1:1000, Santa Cruz Biotechnology, cat. # sc-47778). Membranes were washed extensively in TBS-T, followed by incubation for 2 h at room temperature with a secondary antibody, either goat anti-mouse (1:10,000, Millipore-Sigma, cat/# AP130P) or goat anti-rabbit (1:10,000; Millipore-Sigma, cat. # AP156P). Finally, membranes were washed in TBS-T, exposed to Luminata™ Classico Western HRP Substrate (Millipore-Sigma) and imaged using a ChemiDoc™ Touch Imaging System (Bio-Rad). Densitometry was performed using ImageLab version 5.2 software (Bio-Rad).

Samples for *O*-GlcNAc immunodot blot analysis were prepared using a Bio-Dot^®^ Microfiltration apparatus (Bio-Rad), as described previously [17]. Briefly, 4 μg of protein was loaded into each well of the dot blot apparatus and then transferred to the nitrocellulose membrane by gravity filtration for 1.5 h. All subsequent steps were identical to the Western blot procedure with the RL2 primary antibody to detect global levels of *O*-GlcNAc in cells.

### 2.3. Quantitative RT-PCR

Total RNA was collected from cells using the Qiashredder (Qiagen, cat. # 79654) and RNeasy Mini kit (Qiagen, cat. # 74104) and was reverse transcribed according to manufacturer’s instructions into cDNA using a High-Capacity cDNA Reverse Transcription Kit (ThermoFisherScientific, cat. # 4368814). PCR reactions containing 500 nM of each forward and reverse primers (Table 1), SensiFAST SYBR Mix (FroggaBio, cat. # BIO-98005), and 1 μL of cDNA were carried out using the CFX Connect Real-Time PCR Detection System (Bio-Rad). The comparative cycle threshold method (2^−ΔΔCt^) was used to determine fold changes in gene expression. The cycle threshold value was normalized to the housekeeping gene *L14* and was subsequently normalized to DMSO-treated controls for F9 cell experiments or to untreated E14 cells for ES and XEN experiments.

### 2.4. RNA Sequencing

RNA was extracted from ESC and XEN cells are described above using Qiashredder and RNeasy kits (both from Qiagen). Samples were quantified, and RNA purity was evaluated using both a Nanodrop spectrophotometer (ThermoFisherScientific) and an Agilent 2100 bioanalyzer (Agilent Technologies). The BGISEQ-500 platform was used for library construction and sequencing, where reads were mapped to reference genome (GRCm38-mm10) using Bowtie [33]. Gene expression levels were then calculated using RSEM [34] and DEseq2 [35] was using to determine differentially expressed genes with a fold change greater than or equal to 2.0 and a significance value less than or equal to 0.0001.

### 2.5. Galectin-3 ELISA

Extracellular galectin-3 was detected using a Mouse SimpleStep ELISA^®^ kit (Abcam, cat. # ab203369). Briefly, supernatants were collected after 3 days of cell culture and centrifuged at 2000× *g* for 10 min, and subsequent steps were carried out according to manufacturer’s instructions. Data analysis was performed after obtaining a standard curve. The secretion levels of galectin-3 from E14 and E4 cells were normalized to intracellular protein concentrations, which were determined using a *DC* Protein Assay kit (Bio-Rad).

### 2.6. Statistical Analysis

All figures are representative of at least 3 independent biological replicates. An unpaired Student’s *t*-test was performed for data comparisons between two groups. One sample *t*-test was performed for data comparisons between two groups where one group was set as a relative value of 1. One-way ANOVA was performed with Tukey’s Honest Significant Difference post-hoc analysis for data comparisons between 3 or more groups. All statistical analyses were performed using GraphPad Prism version 8.4.2, (GraphPad Software, San Diego, CA, USA). *P*-values were considered statistically significant at * *p* < 0.05, ** *p* < 0.01, *** *p* < 0.001, and **** *p* < 0.0001.

## 3. Results

### 3.1. Global Reduction in O-GlcNAcylation Levels in Feeder-Free XEN Cells

Global *O*-GlcNAcylation levels were determined in feeder-free mouse E14 embryonic stem (ES) cells and terminally differentiated mouse E4 extraembryonic endoderm (XEN) cells. Western blot analysis showed multiple *O*-GlcNAcylated proteins in both cell lines with a significant decrease in global levels of *O*-GlcNAc in the XEN cells as quantified by immunodot blot technique (Figure 1a). *O*-GlcNAcylation status was also examined by measuring the expression of *Ogt*, which encodes the enzyme that catalyzes the addition of GlcNAc to serine/threonine residues, and the expression of *Oga*, which encodes the enzyme responsible for removing the modification. *Ogt* and *Oga* expression levels were found to be significantly decreased in XEN cells, but the ratio of *Ogt/Oga* was found to be significantly higher in XEN cells (Figure 1b). At first glance, this increased *Ogt/Oga* ratio would suggest more *O*-GlcNAc accompanied differentiation; however, it contradicts the dot blot analysis used to examine global *O*-GlcNAc levels (Figure 1a). Further evidence supports the global decrease in *O*-GlcNAc observed in Figure 1a, as the expressions of the rate-limiting enzyme of the hexosamine biosynthesis pathway, *Gfpt1* and *Gfpt2*, both show significantly reduced expression in XEN (Figure 1c). This decrease in *O*-GlcNAcylation with differentiation was also representative in the F9 cell model (Appendix A) and by previous reports from this lab [17] and others [15,18]. Incidentally, F9 cells differentiated toward XEN showed increased expression in *Ogt* and *Oga,* but there was no apparent change in the *Ogt/Oga* ratio (Appendix A).

### 3.2. OGA Inhibition Alters Pluripotency and XEN Marker Expression Patterns

To determine the effects of global *O*-GlcNAcylation levels on pluripotency and endodermal markers, ES and XEN cells were treated with 10 μM Thiamet G (TG), a well-established OGA inhibitor elevating *O*-GlcNAc levels in different types of cells [17,36]. As an example, the efficiency of this treatment was evident in XEN cells, which originally had relatively low homeostatic level of *O*-GlcNAc (Figure 2a). The inhibition of OGA for 24 h in ES cells resulted in no significant change in *Oct4* expression (Figure 2b), but there was decreased *Nanog* expression (Figure 2b). Exploring the expression of XEN markers *FoxA2*, *Gata6*, and *Dab2* [31,37] in TG-treated ES cells all showed no significant changes (Figure 2b). Furthermore, OGA inhibition in XEN cells resulted in no significant change in *Gata6* expression, but caused a decrease in *FoxA2* and *Dab2* levels (Figure 2c). Surprisingly pluripotency marker *Nanog* had reduced expression in TG-treated XEN cells; however, there was no change in *Oct4* (Figure 2c). Although the decrease in *FoxA2* and *Dab2* in XEN cells would suggest that the high global *O*-GlcNAc might positively regulate pluripotency, the simultaneous decrease in *Nanog* expression in ES cells and XEN following the chemically induced inhibition of OGA contradicts this idea.

### 3.3. Lgals3 Expression Is Decreased in XEN Compared to ES Cells

Since we and others have shown that global *O*-GlcNAcylation decreased in differentiated cells [15,17,18] and our observed increase in *O*-GlcNAcylation with TG treatment would be expected to favor pluripotency, the overall levels of *O*-GlcNAcylated proteins should also decrease with differentiation. We previously demonstrated altered oxidative stress during XEN-like cell differentiation [26] and have shown that the expression and levels of galectins are affected in response to differentiation and oxidative stress in HL-60 cells [38]. Galectins are proteins presumably regulated by *O*-GlcNAcylation, which impacts their intracellular and extracellular functions [21]. RNA sequencing analysis of transcripts from ES and XEN cells was employed to explore general trends in the expression of galectin genes and others involved in the *O*-GlcNAcylation process (*Ogt*, *Mgea5*, *Gfpt1* and *Gfpt2*). Our sequencing analysis showed changes in many galectin genes, with some such as *Lgasl3* showing decreased transcript levels in XEN cells and others such as *Lgals2* showing increased transcript levels (Figure 3a). Although RNA sequencing analysis showed altered expression levels of various galectin genes, validation by RT-qPCR revealed that all but *Lgals3* (Figure 3b), encoding galectin-3 protein, showed no difference in expressions between ES and XEN cells or had extremely variable expression between replicates, making the statistical analysis of these genes less reliable (Figure 3b). In contrast, F9 cells showed a significant decrease in *Lgals1*, *Lgals2*, *Lgals4*, *Lgals7*, *Lgals8*, *Lgals9*, and *Lgals12*, but no significant change in *Lgals3* expression with differentiation (Appendix A) was observed, which again underscores the differences between E14 and F9 cell types. As the functional role of galectins, which are soluble proteins, may depend not only on their expression levels but also on their *O*-GlcNAc-mediated localization in cells [21,22,23,24], we next analyzed intracellular levels of galectin-3 and galectin-3 secretion in the context of embryonic stem cell differentiation.

### 3.4. Elevated Extracellular to Intracellular Ratio of Galectin-3 in XEN Cells

Galectin-3 is a candidate protein, the secretion of which is regulated by *O*-GlcNAcylation [22]. Our Western blot analysis showed a significant decrease in intracellular galectin-3 in XEN compared to ES cells (Figure 4a), as did an ELISA examining the extracellular levels of galectin-3 (Figure 4b). However, the ratio of extracellular to intracellular levels of galectin-3 significantly increased in XEN versus ES cells (Figure 4c), indicating that although there is less intracellular galectin-3 overall, XEN cells were prone to secrete this galectin. Thus, XEN cells had less *Lgals3* expression, less overall galectin-3 levels, and more intensive galectin-3 secretory activity, which accompanied extraembryonic endoderm differentiation. Therefore, since the localization of galectins influences their function [19], this extracellular shuttling or secretion of galectin-3 may be due to reduced global *O*-GlcNAcylation. The changing localization of galectin-3 highlights its highly likely role during differentiation and highlights an intracellular function when cells are in the pluripotent state.

### 3.5. Inhibiting OGT in ES Cells Increases Galectin-3 Secretion

To explore whether low *O*-GlcNAcylation is a driving force of galectin-3 secretion, we tested the effects of an OGT inhibitor alloxan [39] on ES and XEN cells. As expected, alloxan at a concentration of 5 mM was effective in reducing global *O*-GlcNAcylation levels in ES cells (Figure 5a). This treatment led to a significant increase in galectin-3 secretion by ES cells as measured by ELISA, while no significant change was evident following OGA inhibition with TG (Figure 5b). Interestingly, no change was seen when XEN cells were treated with either alloxan or TG (Figure 5b), suggesting an irreversibly upregulated galectin-3 secretory system in this case. Thus, these results would suggest that galectin-3 localization in ES cells is dependent on *O*-GlcNAc homeostasis, as noted earlier [22], and this localization may serve as an indicator to discriminate pluripotency from differentiation.

### 3.6. OGA Inhibition Is Not Sufficient to Inhibit XEN Differentiation

Decreased global *O*-GlcNAc levels in XEN cells (Figure 1a) coincided with the decrease in intracellular galectin-3 levels (Figure 4a) and increased the secretion of galectin-3 (Figure 4c), which was also stimulated by an OGT inhibitor alloxan in ES cells (Figure 5b). Thus, it is possible that *O*-GlcNAc and the localization of galectin-3 could influence pluripotency. If so, inhibiting OGT activity would promote differentiation while inhibiting OGA, which we showed previously (Figure 2), would promote pluripotency. To test this further and examine over time what effect *O*-GlcNAcylation had on pluripotency and differentiation, ES cells were differentiated toward XEN lineage for 8 days in the presence or absence of TG to disrupt OGA activity (Figure 6). The regular differentiation of ES cells toward a XEN lineage was confirmed through morphology as ES cells form tight colonies, while XEN cells and ES cells were induced to differentiate show a cobblestone-like morphology with some rounded cells (Figure 6a). Cell differentiation was also confirmed by conducting gene expression analysis demonstrating that pluripotency markers *Oct4* and *Nanog* have reduced expression, while differentiation markers *Gata6* and *Dab2* increased in expression over time (Figure 6b). Our results from testing differentiation in the presence of TG revealed that the relative expression of the pluripotency marker *Nanog* decreased similarly to differentiation conditions in the absence of TG (Figure 6c). In addition, *Oct4* expression in the presence of TG also decreased to the same extent as differentiation conditions in the absence of TG (Figure 6c). This suggests that the inhibition of OGA did not promote pluripotency. Furthermore, blocking OGA did not alter the expression of differentiation markers *Gata6* and *Dab2* (Figure 6c), which again contradicts the predictions. Together, this evidence would suggest that the inhibition of OGA, which increases global *O*-GlcNAcylation in stem cells (Figure 2), neither promotes pluripotency nor inhibits differentiation.

## 4. Discussion

The objective of this study was to understand the role of *O*-GlcNAcylation during XEN differentiation and to explore if this specific post-translational modification can regulate galectin expression and localization. Here, we report that global *O*-GlcNAc decreased in XEN cells, although inhibiting OGA had no effect on differentiation. Galectin-3 localization and secretion, however, depend on both cell differentiation status and *O*-GlcNAc homeostasis.

Although the reduction in global *O*-GlcNAcylation is associated with the differentiation programs of many cell lineages, there are a few exceptions [21]. In our model, a decrease in global *O*-GlcNAc in XEN compared to ES cells (Figure 1) is consistent with previous work on neutrophilic and enterocytic differentiation [17] and supports another study that showed decreased global *O*-GlcNAcylation when ES cells were differentiated toward cardiomyocytes [18]. The global decrease in *O*-GlcNAcylation, as seen during XEN and cardiomyocyte differentiation, coincided with the decreased expression of *Ogt,* the gene encoding the enzyme responsible for adding *O*-GlcNAc to proteins (Figure 1) [18]. This overall decrease in *O*-GlcNAcylation is likely in response to the exit from pluripotency, and this was observed by Jang et al. who demonstrated the pluripotency markers OCT4 and SOX2 are both modified by *O*-GlcNAc, which is later removed during differentiation [15].

Since the trend of a high degree of the *O*-GlcNAcylation is consistent with pluripotency and decreases occur in response to differentiation, we explored how chemical inhibition of the modification-removing enzyme, OGA, would affect both stem cells and those induced to form XEN. Blocking OGA was expected to increase the amount of global *O*-GlcNAc, and this was expected to maintain or enhance pluripotency in ES cells, while decreasing the differentiation to XEN. In fact, OGA inhibition in XEN cells did decrease XEN marker *FoxA2* expression (Figure 2e); however, it did not inhibit the differentiation of ES cells induced toward XEN lineage (Figure 6). In contrast, OGA inhibition in ES cells did inhibit the expression of the pluripotency marker *Nanog* (Figure 2c); however, it did not inhibit the differentiation-induced decrease in pluripotency markers in ES cells induced toward XEN (Figure 6).

Previous work has provided evidence that proteins within the galectin family can be regulated by *O*-GlcNAcylation [17,21,22]. Few studies have examined galectins in F9 cells [40,41], where we saw no changes in *Lgal3* expression in F9 cell XEN differentiation (Appendix A); however, we observed a decrease in *Lgals3* expression in terminally differentiated XEN cells compared to ES cells (Figure 3). Unlike the changes in *O*-GlcNAc levels, which are consistent in many previous reports [15,17,18,21], galectins appear to have context-specific changes in expression, as evident from our own studies [17,20,38].

As *Lgals3* was the only galectin for which its expression decreased in XEN cells compared to ES cells (Figure 3b), we further explored the regulation of this protein in pluripotency and differentiation. Similarly to other members of the galectin family, galectin-3 is a multifunctional molecule with different modes of action inside and outside the cell [19]. In addition to confirming previous findings of low intracellular galectin-3 expression in XEN cells [37], ELISA analysis revealed a significant increase in galectin-3 secretion in this cell model (Figure 4). Although it is not explored in this study, others have reported on the role of galectin-3 in clathrin-independent endocytosis by interacting with secreted and membrane bound glycoproteins [19,22], which could alter detectable galectin-3 through ELISA. In a developmental context, Iacobini et al. discussed the role of galectin-3 in osteogenic lineages, highlighting its localization as enhancing or inhibiting specific cell types [42]. Given our results showing increased intracellular galectin-3 in ES cells, it is tempting to speculate that extracellular galectin-3 is playing a role in enhancing XEN differentiation. Firstly, the alloxan studies, through the inhibition of OGT, reduced the extent of *O*-GlcNAcylation (Figure 5a) and increased extracellular galectin-3 levels in ES cell culture (Figure 5b), which is in line with the study of Mathew et al. who showed that the enhanced secretion of galectin-3 occurs as a result of it not being tagged with *O*-GlcNAc [22]. Although used in the present study as an OGT inhibitor, a limitation of alloxan is its unspecific inhibition of OGT that can also, at high concentrations, inhibit OGA [43]. Given this previous study, we chose an alloxan concentration that was unlikely to reduce OGA activity from controls.

Our findings support the new concept of the *O*-GlcNAc-mediated regulation of galectin localization and secretion in cells [21]. Galectin-3 secretion is considered to occur through unconventional mechanisms, meaning that it is not released from the cell through the classical endoplasmic reticulum/Golgi secretory pathway [19]. Instead, other means include direct translocation, export via the lysosome or endosome, export via microvesicles, and/or release in exosomes [44,45,46]. *O*-GlcNAcylation is known to regulate protein sorting exosomes; however, it is still unclear if its regulation is binary or a more dynamic process [47]. To our knowledge, no studies have investigated the composition or role of exosomes in XEN differentiation, so galectin-3 may be exiting the cell regardless of its *O*-GlcNAc status. While this requires further study, our evidence indicates that the decrease in global *O*-GlcNAc status occurs concurrently with increased galectin-3 secretion and the exit from pluripotency.

Given the differences in the two populations, ES and XEN cells, we wanted to explore the role of OGA inhibition when ES cells were induced toward the XEN lineage. Given that OGA inhibition increased global *O*-GlcNAc and decreased *FoxA2* expression in XEN cells (Figure 2), it was predicted OGA inhibition should inhibit XEN differentiation. However, when ES cells were induced toward XEN in the presence of the OGA inhibitor, there was no apparent change in pluripotency and XEN marker expression compared to differentiating cells in the absence of the inhibitor (Figure 6). As others have demonstrated the importance of increased *O*-GlcNAc in ES cell self-renewal and the maintenance of pluripotency transcription factor function [15], this result was unexpected. For instance, Maury et al. demonstrated that excess *O*-GlcNAc inhibited human pluripotent stem cell differentiation without inhibiting cell exits from pluripotency [16]. Taken together, this suggests that although there are changes in *O*-GlcNAc globally in XEN cells compared to ES cells, inhibiting only the enzyme involved in removing *O*-GlcNAc modifications may not be sufficient to inhibit XEN differentiation. The ultimate role of *O*-GlcNAc in ES/XEN model is still undergoing evaluation, and other cell models and experimental design may help to further elaborate this notion.

Despite some conflicting information, which is likely cell type-dependent, we were the first to explore the role of *O*-GlcNAcylation in XEN maintenance and differentiation. Our work highlighted the differences in global *O*-GlcNAc between ES and XEN cells and showed that galectin-3 intracellular levels decreased in XEN; more importantly, the elevation of galectin-3 secretion was observed as the result of *O*-GlcNAcylation occurring concomitantly with XEN differentiation.

## Figures and Tables

**Figure 1 biomolecules-12-00623-f001:**
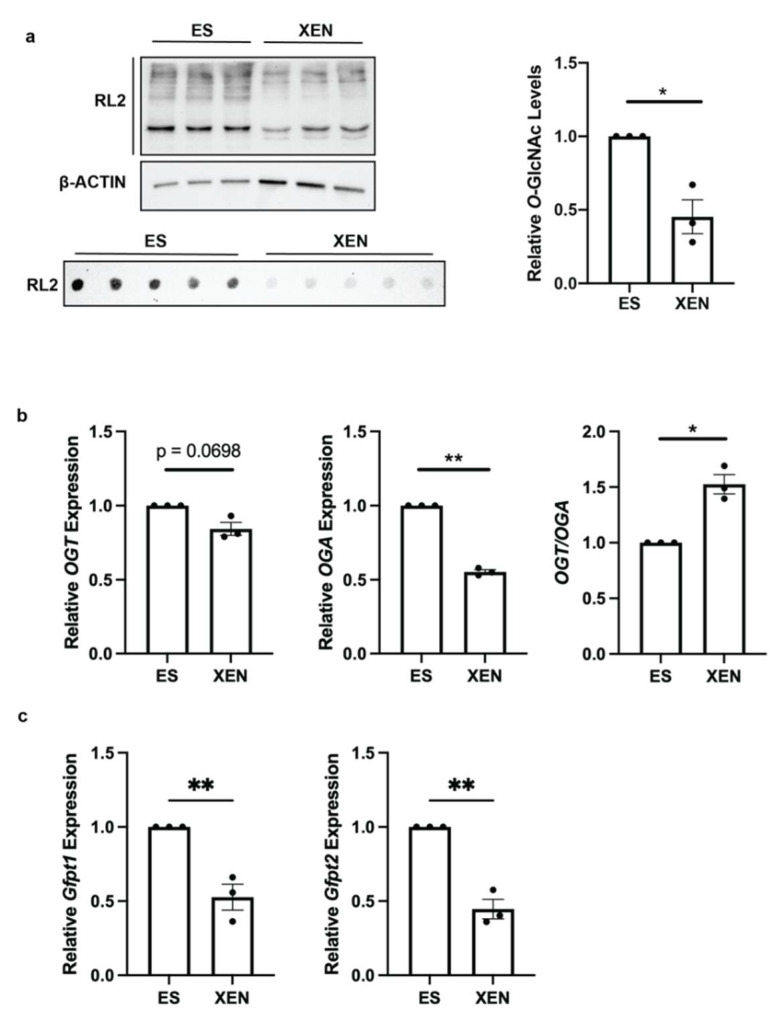
Global *O*-GlcNAcylation is decreased in XEN compared to ES cells: (**a**) Western blot, immunodot blot, and immunodot blot densitometry of *O*-GlcNAcylated proteins from ESC and XEN cells detected by the RL2 antibody; (**b**) expression levels and ratio of *Ogt* and *Oga* transcripts and (**c**) expression levels of *Gfpt1* and *Gfpt2* in ES and XEN cells as detected by RT-qPCR. Bars represent mean ± SEM, N = 3. * *p* < 0.05, ** *p* < 0.01.

**Figure 2 biomolecules-12-00623-f002:**
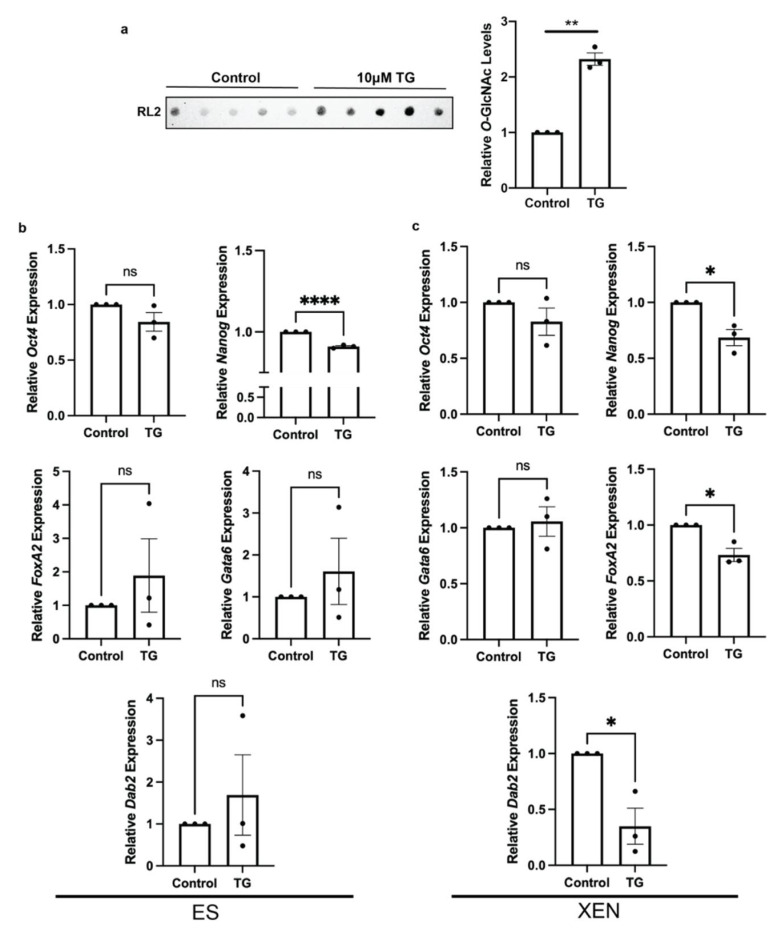
OGA inhibition alters pluripotency marker expression in ES cells and XEN marker expression in XEN cells. The cells were treated with 10 μM TG for 24 h: (**a**) immunodot blot and densitometry of *O*-GlcNAcylated proteins detected by the RL2 antibody in XEN cells; (**b**,**c**) expression of pluripotency gene expression *Oct4* and *Nanog* and XEN marker expression *FoxA2, Gata6*, and *Dab2* in ES cells (left panel) and XEN cells (right panel). Bars represent mean values ± SEM, N = 3. * *p* < 0.05, ** *p* < 0.01, **** *p* < 0.0001.

**Figure 3 biomolecules-12-00623-f003:**
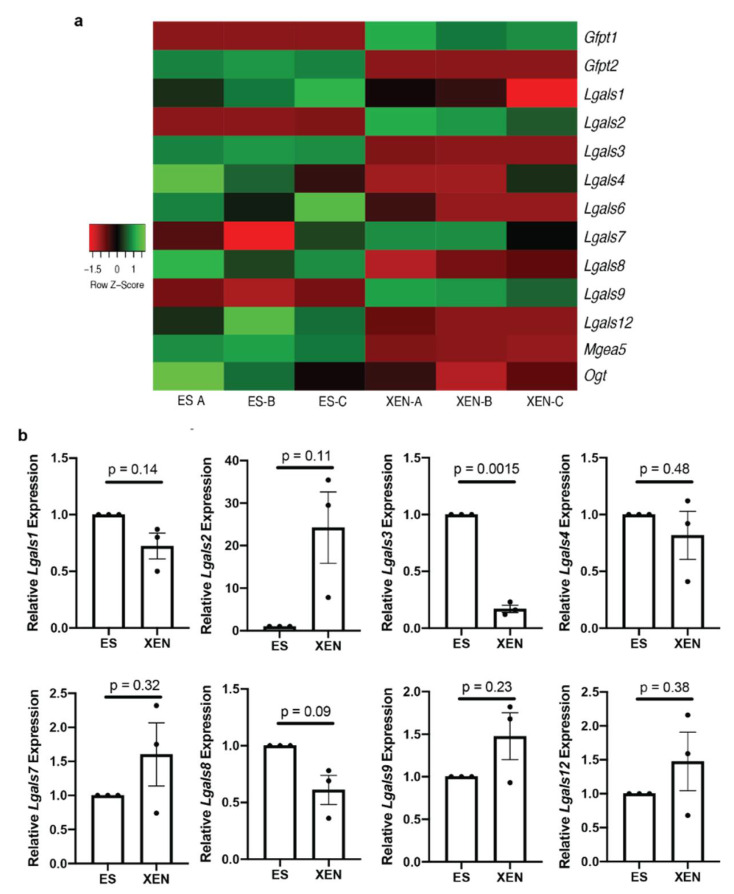
RNA sequencing and gene expression analysis of galectin genes in ES cells and XEN: (**a**) heat map of RNA sequencing of transcripts in ES cells and XEN showing *Lgals* genes 1–4, 7–9, and 12 and genes involved in *O*-GlcNAcylation *Ogt*, *Mgea5*, *Gfpt1,* and *Gfpt2*; data are shown for 3 biological replicates denoted as -A, -B, and -C; (**b**) expression levels of *Lgals1*, *Lgals2*, *Lgals3*, *Lgals4*, *Lgals7*, *Lgals8*, *Lgals9*, and *Lgals12* as detected by RT-qPCR. Bars represent mean ± SEM, N = 3.

**Figure 4 biomolecules-12-00623-f004:**
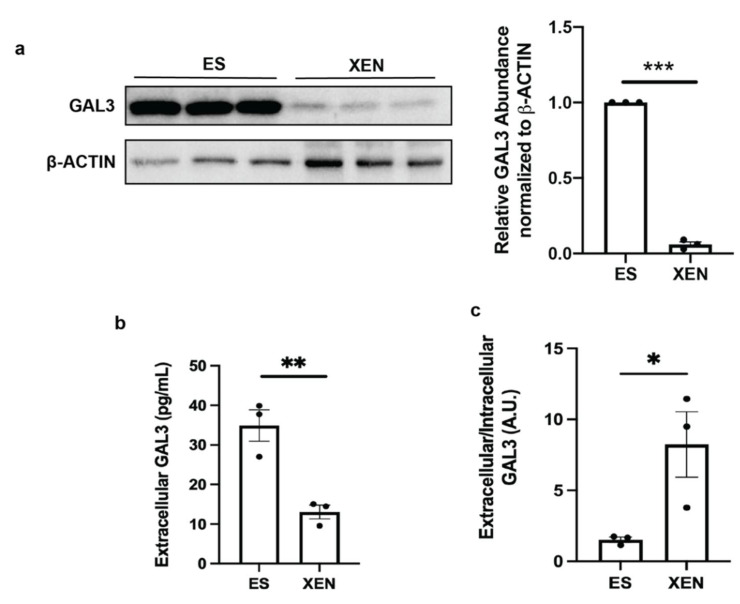
ES cells increased intracellular galectin-3 levels while XEN cells increased extracellular galectin-3 levels compared to their own intracellular protein levels: (**a**) Western blot and densitometry of intracellular galectin-3 in ES and XEN cells; (**b**) ELISA of extracellular galectin-3 in the cell culture media of ES and XEN cells after 8 days normalized to total intracellular protein levels (mg/mL); (**c**) extracellular to intracellular ratio of galectin-3 in ES and XEN cell cultures both normalized to total intracellular protein levels (mg/mL) represented as arbitrary units (A.U.). Bars represent mean ± SEM, N = 3. * *p* < 0.05, ** *p* < 0.01, *** *p* < 0.001.

**Figure 5 biomolecules-12-00623-f005:**
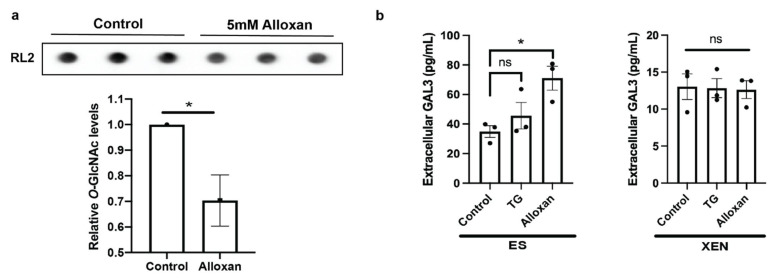
OGT inhibition in ES cells increases extracellular galectin-3. (**a**) Immunodot blot of *O*-GlcNAc detecting antibody RL2 in ES cells treated with 5 mM alloxan, an OGT inhibitor, and densitometry; (**b**) ELISA measuring secretion of galectin-3 from ES and XEN cells treated with 10 μM TG or 5 mM alloxan for 8 days normalized to total intracellular protein (mg/mL). Bars represent mean values ± SEM, N = 3. * *p* < 0.05.

**Figure 6 biomolecules-12-00623-f006:**
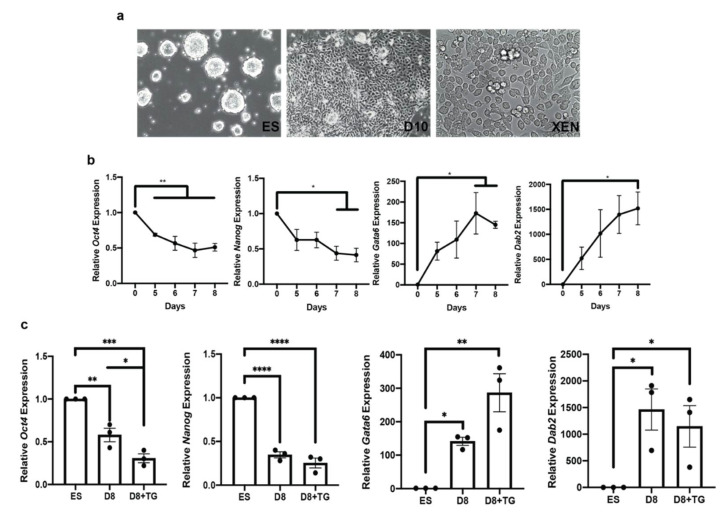
OGA inhibition is not sufficient to inhibit ESC differentiation toward XEN phenotype: (**a**) phase contrast microscopy of undifferentiated ES, differentiated ES toward XEN after 10 days in differentiation media, and XEN cells; (**b**) time-dependent changes in the expression of genetic biomarkers of pluripotency (*Oct4* and *Nanog*) and differentiation (*Gata6* and *Dab2*) in ES cells differentiated toward the XEN phenotype; (**c**) effects of TG (10 μM) on the expression of same biomarkers of pluripotency/differentiation as in (**b**) on day 8 of ES cell differentiation toward XEN. Bars represent mean ± SEM, N = 3. * *p* < 0.05, ** *p* < 0.01, *** *p* < 0.001, **** *p* < 0.0001.

**Table 1 biomolecules-12-00623-t001:** Primers used for RT-qPCR.

Gene	Forward Primer (5′→3′)	Reverse Primer (5′→3′)
*Lgals1*	TCTCAAACCTGGGGAATGTCTC	CTCAAAGGCCACGCACTTAATC
*Lgals2*	AACATGAAACCAGGGATGTCC	CGAGGGTTAAAATGCAGGTTGAG
*Lgals3*	AGGAGAGGGAATGATGTTGC	TAGCGCTGGTGAGGGTTATG
*Lgals4*	GGTCGTGGTGAACGGAAATTC	GTGGAGGGTTGTACCCAGGA
*Lgals7*	GTGAGGAGCAAGGAGCAGAT	CGGTGGTGGAAGTGGAGATA
*Lgals8*	CCGATAATCCCCTATGTTGG	GTTCACTTTGCCGTAGATGC
*Lgals9*	TTGAGGAAGGAGGGTATGTG	AACTGGACTGGCTGAGAGAA
*Lgals12*	TATGGCACAACAATTTTTGGTGG	GCTTGACAGTGTAGAATCGAGGG
*Ogt*	TTGGCAATTAAACAGAATCCCCT	GGCATGTCGATAATGCTCGAT
*Oga*	TGGTGCCAGTTTGGTTCCAG	TGCTCTGAGGTCGGGTTCA
*Gfpt1*	GAAGCCAACGCCTGCAAAATC	CCAACGGGTATGAGCTATTCC
*Gfpt2*	CCAACGGGTATGAGCTATTCC	GACTCTTTCGACCAATGTGGAA
*Oct4*	CCCAATGCCGTGAAGTTGGA	GCTTTCATGTCCTGGGACTCCT
*Nanog*	TCTTCCTGGTCCCCACAGTTT	GCAAGAATAGTTCTCGGGATGAA
*FoxA2*	CCCTACGCCAACATGAACTCG	GTTCTGCCGGTAGAAAGGGA
*Dab2*	GGAGCATGTAGACCATGATG	AAAGGATTTCCGAAAGGGCT
*Gata6*	ATGGCGTAGAAATGCTGAGG	TGAGGTGGTCGCTTGTGTAG
*L14*	GGGAGAGGTGGCCTCGGACGC	GGCTGGCTTTCACTCAAAGGCC

## Data Availability

A data appears in this manuscript.

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
