# Peer review of "O-GlcNAcylation and Regulation of Galectin-3 in Extraembryonic Endoderm Differentiation"

_biomolecules, 2022, doi:10.3390/biom12050623_

Round 1

Reviewer 1 Report

The authors here explore the expression of the post-translational modification O-linked GlcNAc on intracellular and nuclear proteins.  They examine this in embryonic stem (ES cells) and extraembryonic endoderm (XEN) cells during differentiation.  Because of the roles of O-GlcNAc in regulating transcription through regulation of transcription factors, they explored whether there are accompanying changes in expression of the extracellular form of galectin-3, a soluble galactose-binding lectin, previously shown to be regulated in part by O-GlcNAc modifications.  Their results, using immunoblots, suggest that overall O-GlcNAc levels decrease in XEN cells relative to ES cells, consistent with decreased expression of OGT, the enzyme that adds O-GlcNAc.  The also observed that the expression of galectin-3 protein, but not the transcript, increased in XEN compared to ES cells, which was modulated by inhibiting O-GlcNAc removal through inhibition by the drug alloxan.  The results suggest an interesting potential relationship between galectin-3 expression, and O-GlcNAc levels, but the authors found no significant effects of inhibiting O-GlcNAc on ability of ES cells to differentiate.  Overall, the studies are well done, and the results are generally d interesting, but some statements and conclusions are confusing.    

There are a few issues the authors need to address.

  • While they examined specifically for other galectins, it appears from Fig. 3 that transcripts of several galectins were significantly modified upon differentiation. The data show a large decrease in Galectin-3 expression upon differentiation from ES to XEN, and expression of several other galectins also changed.  But the authors appear to make the contradictory statement that “These changes observed in galectin genes were validated by RT-qPCR, which showed that all but Lgals3 (Fig. 3b), encoding galectin-3 protein, showed no difference in expression between ES and XEN cells (Fig. 3b).”  They also state that “As Lgals3 was the only galectin whose expression decreased in E4 XEN cells com-355 pared to E14 ES cells (Fig. 3b)”, but that figure is confusing. The replicates for expression for Galectin-3 seems tight, but for many others the replicates were scattered (see for example galectins-2 and 9), thus to this reviewer, it is not possible to calculate the significance. 

  • The authors did not examine protein expression other than through ELISA, but could this be affected by binding to ligands? Perhaps secreted glycoproteins or perhaps intracellular non-carbohydrate ligands, as proposed by some investigators?    

  • The statement “Our Western blot analysis showed a significant decrease of both galectin-3 intracellular and extracellular levels in XEN cells compared to ES cells (Fig. 4a and b)”, seems to be at odds with the statement in the abstract that “an ELISA revealed increased extracellular galectin-3 in XEN compared to ES cells, which secreted more galectin-3 in response to O-GlcNAc transferase inhibition”. What they observed was differences in location of galectin-3, not expression, but this could be better worded and explained.

Reviewer 2 Report

The manuscript is interesting and the experiments are well designed. However, the following points should be addressed:

  1. Changes in the global level of O-GlcNAC have been associated with Galectin-3 secretion levels-- the potential mechanism is missing.
  2. Can the authors justify, how the RNA sequencing data is helping more than qRT-PCR to delineate any mechanism.
  3. Authors should analyze the O-GlcNAC proteome in ES and XEN cells and figure out which proteins are getting modified--this experiment will be crucial to understand the mechanism. However, if the author can propose a potential mechanism / regulation through any other targeted protein modification; then that should also suffice the mechanistic aspect of this purpose.

Reviewer 3 Report

In this paper from Gatie et al, the authors investigated the potential variations of O-GlcNAcylation levels occurring during extraembryonic endoderm (XEN) differentiation. In parallel, they also analyzed the expression and secretion of galectin-3, a previously identified O-GlcNAcylated protein, in XEN cells compared to embryonic stem (ES) cells. They observed a global decrease in O-GlcNAcylation levels in XEN cells in comparison with ES cells, correlated with a decreased expression of galectin-3 and an increase in its secretion. Then they investigated the effects of O-GlcNAcylation levels modulations (either through inhibition of OGT with alloxan or OGA inhibition with thiamet G) on galectin-3 expression and XEN differentiation. They thus demonstrated that OGT inhibition reduces galectin-3 secretion in ES cells but not in XEN cells. However, their results also showed that OGA inhibition has no effect neither on galectin-3 secretion nor in pluripotency and XEN differentiation. These very preliminary results, although suggesting that O-GlcNAcylation may regulate galectin-3 secretion, tend to show that it is not essential for the differentiation of the extra-embryonic endoderm.

Overall, the data presented here sometimes lack some controls and require additional analysis of the O-GlcNAcylation machinery to support their conclusions. Furthermore, the OGT inhibition experiments were conducted with alloxan, a non-specific inhibitor of the enzyme while other specific inhibitors are commercially available. Therefore, I recommend additional experiments so that the paper can be accepted.

- in the experiments presented in Figure 1, the authors observed a discrepancy between ogt/oga mRNAs ratio and O-GlcNAcylation levels in XEN cells. However, mRNAs levels do not always reflect protein expression. A western Blot analysis of OGT and OGA expression should be performed to definitively conclude. Furthermore, on the heat map of the figure 2a, one can see that the genee expression of the two isoforms of GFAT (gfpt1 and gfpt2), the rate limiting enzyme of the Hexosamine Biosynthesis Pathway that provides UDP-GlcNAC, is also modulated in XEN cells. This result should be confirmed by RT-qPCR and Western Blot analysis.

- Figure 2: The authors present contradictory results between the expression of pluripotency/differentiation markers in response to thiamet G treatment but they compared the results of qPCR analysis of pluripotency markers performed in ES cells (figure 2b,c) with the analysis of markers of differentiation in XEN cells (fig 2d) . The analysis of the four selected markers should be performed in the same cell type. Moreover, could the authors explain why they choose to analyze specifically the expression of gata6 and FoxA2 and their link with pluripotency/differentiation? Is there any other markers of XEN differentiation that could be studied? Furthermore, in figure 2a, they only show immunodot blots of O-GlcNAcylated proteins for the XEN cells. What about ES cells?

-Figure 5: To support the conclusions drawned in the figure 5b, immunodotblot analysis of the thiamet-G treated ES and XEN cells and of the Alloxan-treated XEN cells that proves the efficiency of the OGT or OGA inhibition should be added. More importantly, as mentioned above, Alloxan is an uracil analog that is not specific to OGT inhibition as it has been demonstrated to also inhibit OGA (ref 38 of the paper). Moreover, it is unstable in solution and can be toxic via the production of ROS. These experiments should be repeated using other more specific and selective inhibitors such as OSMI-4 or As5S-GlcNAc

- From the experiments presented in figure 6, the authors conclude that OGA inhibition is not sufficient to inhibit the differentiation of ES cells towards a XEN phenotype. However, the authors did not analyze the variations of O-GlcNAcylation and its machinery in this differentiation model. So, is this model relevant?

Round 2

Reviewer 1 Report

The manuscript has been revised to address the criticisms raised, and it is improved. 

Author Response

Thank you for the comments and endorsement.

Reviewer 3 Report

In its revised form, the authors have addressed most of my comments, however some issues still need to be clarified before the manuscript can be published.

I don’t agree with the authors when they say that “…with results in Fig.1a, where high levels of O-GlcNAc were seen in ES cells, we were confident that performing the immunodot blot of ES cells treated with TG would not show any detectable changes in O-GlcNAc levels.” Whatever the basal level of intracellular O-GlcNAcylation, inhibition of OGA should nonetheless lead to an increase in o-GlcNAcylation levels. Similarly, inhibition of OGT in XEN cells should lead to a decrease in O-GlcNAcylation which is also not shown.

For me, the results concerning the possible involvement of O-GlcNAcylation in pluripotency/differentiation of cells are still unclear. In their first set of experiments the authors compare ES cells and XEN cells. They clearly show a decrease in O-GlcNAcylation and galectin-3 secretion in XEN cells compared to ES cells. They also show that the inhibition of OGA in XEN cells is associated with a decrease of some differentiation markers but on the other hand it is also associated with a decrease of a pluripotency marker in both cell types which, they say, contradicts this idea (lines 223-226). In a second set of experiments, they use a model of ES cell differentiation into an XEN phenotype. In this model, did they also observed a decrease in O-GlcNAcylation levels and galectin-3 secretion after induction (let alone the TG condition)? If not, I still wonder if this model is relevant.

Author Response

Please see attached letter.
